# Obesogenic diet induces circuit-specific memory deficits in mice

**Ioannis Bakoyiannis[1†], Eva Gunnel Ducourneau[1†], Mateo N'diaye[1], Alice Fermigier[1], Celine Ducroix-Crepy[1], Clementine Bosch-Bouju[1], Etienne Coutureau[2], Pierre Trifilieff[1], Guillaume Ferreira[1]\***

[1]University of Bordeaux, INRAE, Bordeaux INP, NutriNeuro, UMR 1286, F-33077, Bordeaux, France; [2]University of Bordeaux, CNRS, INCIA, UMR 5287, 33077, Bordeaux, France

**Abstract** Obesity is associated with neurocognitive dysfunction, including memory deficits. This is particularly worrisome when obesity occurs during adolescence, a maturational period for brain structures critical for cognition. In rodent models, we recently reported that memory impairments induced by obesogenic high-fat diet (HFD) intake during the periadolescent period can be reversed by chemogenetic manipulation of the ventral hippocampus (vHPC). Here, we used an intersectional viral approach in HFD-fed male mice to chemogenetically inactivate specific vHPC efferent pathways to nucleus accumbens (NAc) or medial prefrontal cortex (mPFC) during memory tasks. We first demonstrated that HFD enhanced activation of both pathways after training and that our chemogenetic approach was effective in normalizing this activation. Inactivation of the vHPC–NAc pathway rescued HFD-induced deficits in recognition but not location memory. Conversely, inactivation of the vHPC–mPFC pathway restored location but not recognition memory impairments produced by HFD. Either pathway manipulation did not affect exploration or anxiety-like behaviour. These findings suggest that HFD intake throughout adolescence impairs different types of memory through overactivation of specific hippocampal efferent pathways and that targeting these overactive pathways has therapeutic potential.

**\*For correspondence:**
guillaume.ferreira@inrae.fr

[†]These authors contributed equally to this work

**Competing interest:** The authors declare that no competing interests exist.

## Editor's evaluation

This valuable study explores the neuronal pathways through which adolescent diet influences later memory processes. The data convincingly show that obesogenic high-fat diet during adolescence impairs distinct forms of memory via distinct ventral hippocampal projections to nucleus accumbens v. medial prefrontal cortex. This study will be of interest to those interested in development, diet, metabolism, learning and memory, and the intersection of these factors.

## Introduction

Obesity is a chronic disease critically affecting public health that has been rising tremendously over the last years. Primarily due to an overconsumption of energy-dense food combined with a sedentary lifestyle, obesity is associated with several comorbidities including cardiovascular and metabolic diseases (*Carbone et al., 2013*), but also cognitive disorders (*Francis and Stevenson, 2013*; *Sui and Pasco, 2020*; *Wang et al., 2014*). In particular, memory deficits have been previously reported in obese adults (*Francis and Stevenson, 2013*; *Martin and Davidson, 2014*; *Sellbom and Gunstad, 2012*; *Yeomans, 2017*), but also in obese adolescents (*Khan et al., 2015*; *Nyaradi et al., 2014*; *Øverby et al., 2013*). This is worrisome because the prevalence of obesity during adolescence, a crucial period for brain development (*Andersen, 2003*; *Spear, 2000*), has drastically risen (*Ogden*

*et al., 2016*). It is therefore timely to identify the mechanisms by which adolescent obesity impairs cognitive functions.

Using animal models, we and others have extensively demonstrated the higher vulnerability of adolescence to the effects of obesogenic diet on hippocampal (HPC) function and HPC-dependent memory as compared to adulthood (*Boitard et al., 2012*; *Boitard et al., 2014*; *Del Olmo and Ruiz-Gayo, 2018*; *Glushchak et al., 2021*; *Khazen et al., 2019*; *Morin et al., 2017*; *Murray and Chen, 2019*; *Tsan et al., 2021*; *Valladolid-Acebes et al., 2013*). However, increasing evidence suggest that, rather than homogeneously altering neuronal function, discrete neuronal subpopulations and pathways are particularly vulnerable to dietary manipulations (*Berland et al., 2020*; *Ducrocq et al., 2020*). Accordingly, few studies reported some effects of adolescent obesity on specific functional circuits in humans (*Vega-Torres et al., 2018*) and animal models (*Moreno-Castilla et al., 2018*). These findings led us to the hypothesis that deficits in distinct HPC-dependent memory could result from alterations of discrete HPC pathways and that manipulating such networks could reverse specific high-fat diet (HFD)-induced memory deficits.

We recently showed that object-based memory is impaired in HFD-exposed animals and that silencing of the ventral hippocampus (vHPC) with Designer Receptor Exclusively Activated by Designer Drugs (DREADD) rescued these memory deficits (*Naneix et al., 2021*). We therefore hypothesized that impairments in distinct components of object-based memory, namely recognition and location, in HFD animals could result from alterations of distinct vHPC efferent pathways. We focused on vHPC projections onto the nucleus accumbens (NAc) and the medial prefrontal cortex (mPFC), because they represent two of the main monosynaptic targets of the vHPC (*Britt et al., 2012*; *Cenquizca and Swanson, 2007*; *Ciocchi et al., 2015*; *Gergues et al., 2020*; *Liu and Carter, 2018*), they modulate object-based memory (*Barker et al., 2017*; *Nelson et al., 2011*; *Sargolini et al., 2003*) and we and others previously demonstrated that, in addition to the HPC, periadolescent HFD affects both NAc and mPFC functions (*Ducrocq et al., 2019*; *Labouesse et al., 2013*; *Labouesse et al., 2017*; *Naneix et al., 2017*; *Reichelt et al., 2019*; *Yaseen et al., 2019*). We found that both pathways were hyperactive in HFD animals after object-based learning. Using an intersectional chemogenetic approach to manipulate selectively each pathway, we demonstrated specific beneficial effects of silencing either NAc- or mPFC-projecting vHPC neurons on HFD-induced deficits in either object recognition memory (ORM) or object location memory (OLM), respectively.

## Results

### Targeting vHPC->NAc or vHPC->mPFC pathways

It is well known that the vHPC sends dense projections to both the NAc and the mPFC. We first replicated these data in our conditions using Ai14(RCL-tdT)-D mice injected with an AAV-CaMKII-Cre in the vHPC (*Figure 1—figure supplement 1A–C*). As shown in *Figure 1*, numerous fibres expressing tdTomato were detected in the NAc (*Figure 1A*) and to a lesser extent in the mPFC (*Figure 1B*).

Given these anatomical relationships, we used an intersectional strategy in order to selectively target either the vHPC->NAc or the vHPC->mPFC pathway. To express an inhibitory DREADD (*Armbruster et al., 2007*) only in NAc-projecting vHPC or in mPFC-projecting vHPC cells, an adeno-associated virus (AAV) carrying a floxed hM4DGi receptor (Gi) expression cassette with the reporter mCherry was injected in the vHPC, while a retrograde CAV-2 vector (*Junyent and Kremer, 2015*, *Alcaraz et al., 2018*) carrying the Cre recombinase was injected in the NAc (*Figure 1C*) or in the mPFC (*Figure 1D*). As a result, only vHPC cells projecting to the NAc or to the mPFC were infected by both vectors and therefore expressed Gi. We indeed detected mCherry labelling in the vHPC, particularly in the ventral CA1 and subiculum, following injection in both the NAc and the mPFC (*Figure 1E,F*, *Figure 1—figure supplement 1D,E*).

### Silencing vHPC->NAc or vHPC->mPFC pathways in HFD animals: effect on brain activation

We next assessed the functional outcomes of silencing either vHPC->NAc or vHPC->mPFC pathways in animals exposed to HFD. Mice of comparable body weight were randomly exposed to either CD ($n = 41$) or HFD ($n = 59$). After 8 weeks of diet exposure, they all received dual virus injections with either DREADD Gi or Control mCherry viruses in vHPC and CAV-Cre in either NAc or mPFC (CD Gi

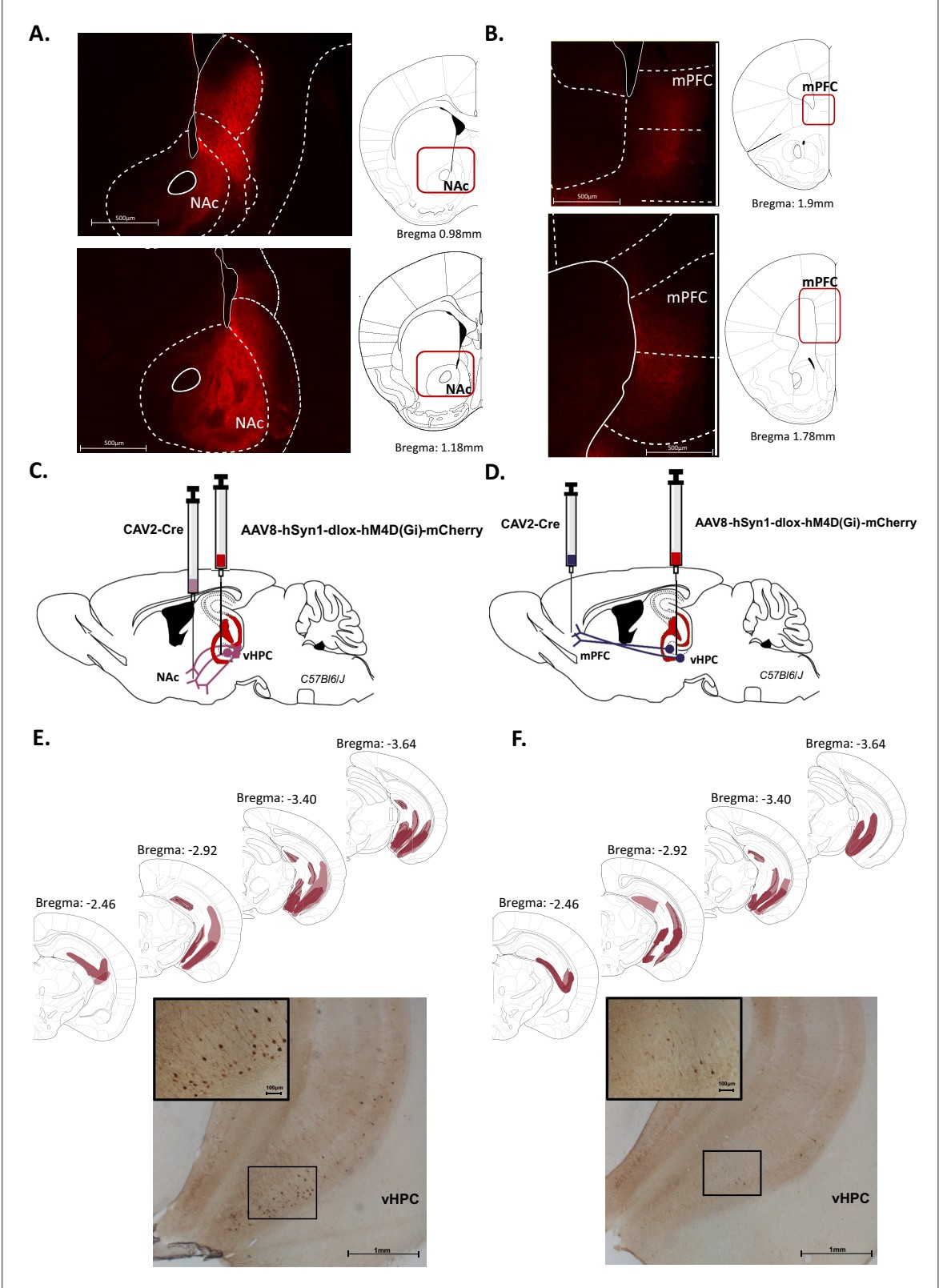

**Figure 1.** Characterization of ventral hippocampus (vHPC) projections to nucleus accumbens (NAc) and medial prefrontal cortex (mPFC). Representative images illustrating expression of TdTomato in fibres in the NAc (**A**) and the mPFC (**B**) after AAV-CaMKII-Cre injection in the vHPC of Ai14(RCL-tdT)-D mice. Schematics adapted from Figures 21 and 23 (**A**) and Figures 14 and 16 (**B**) from *Paxinos and Franklin, 2004*, indicating the levels of fibres labelling. (**C, D**) Schema of intersectional chemogenetic approach. An AAV-hSyn1-dlox-hM4D(Gi)-mCherry vector was injected into the vHPC, while a

*Figure 1 continued on next page*

*Figure 1 continued*

retrograde CAV2-Cre vector was injected in the NAc (**C**) or the mPFC (**D**). (**E, F**) Expression of mCherry is depicted for each condition after amplification using immunohistochemistry. Schematics adapted from Figures 51, 55, 59 and 61 from *Paxinos and Franklin, 2004*, indicating the largest (light red) or the smallest (dark red) viral infection. Representative images illustrating mCherry expression after CAV2-Cre injection in (**E**) the NAc or (**F**) the mPFC. Scale bar is set to 100 μm, 500 μm, or 1 mm.

The online version of this article includes the following figure supplement(s) for figure 1:

**Figure supplement 1.** Labelling in the ventral hippocampus (vHPC).

vHPC–NAc *n* = 8, CD Gi vHPC–mPFC *n* = 9 and CD Control vHPC–NAc/mPFC *n* = 14; HFD Gi vHPC–NAc *n* = 22, HFD Gi vHPC–mPFC *n* = 22, and HFD Control vHPC-NAc/mPFC *n* = 15). Four weeks after surgery (e.g. after 12 weeks of diet exposure), we found that diet exposure was efficient to induce obesity syndrome since there was a significant diet effect on both body weight and fat mass [$F_{(1,82)}$ = 9.8, p = 0.0025 and $F_{(1,82)}$ = 30.0, p < 0.0001, respectively; other comparisons: *F*'s < 2, p > 0.1; see *Table 1*].

We then evaluated the effect of object-based training on c-Fos expression. Mice were exposed to a novel context containing two identical novel objects (*Figure 2A*). CD and HFD groups similarly explored the novel objects excluding any behavioural influences on c-Fos expression between groups (*Table 2*). We first quantified c-Fos-positive cells in CA1–subiculum (*Figure 2B*). As shown in *Figure 2C*, exposure to HFD resulted in an increased c-Fos expression in Control mice as demonstrated by a significant effect of diet [$F_{(1,23)}$ = 6.4, p = 0.019]. Importantly, this increase was also evident when the double stained cells, c-Fos+ and mCherry+, are considered [$F_{(1,21)}$ = 4.5, p = 0.001; *Figure 2D,E*], indicating that HFD exposure increases the amount of activated CA1 cells projecting to either the NAc or the mPFC. Most notably, the pattern of results was different in CA1 when the DREADD groups are considered. Indeed, as shown on *Figure 2C,D* silencing either vHPC->NAc or vHPC–mPFC pathway resulted in an attenuation of c-Fos+ cells. Separate analyses of vHPC–NAc and vHPC–mPFC groups demonstrated, in addition to a main effect of diet [$F_{(1,25)}$ = 4.0, p = 0.056 and $F_{(1,26)}$ = 5.3, p = 0.029, respectively], a main effect of DREADD [$F_{(1,25)}$ = 3.7, p = 0.06 and $F_{(1,26)}$ = 11.0, p = 0.027; interaction: *F* < 2.5, p > 0.1; *Figure 2C*]. Comparable results were obtained when focusing on the double stained cells in both vHPC–NAc and vHPC–mPFC groups since there was a significant effect of diet [$F_{(1,23)}$ = 9.0, p = 0.0063 and $F_{(1,25)}$ = 12.6, p = 0.0016, respectively] and of DREADD [$F_{(1,23)}$ = 8.0, p = 0.0093 and $F_{(1,25)}$ = 10.3, p = 0.0036; interaction: $F_{(1,23)}$ = 3.6, p = 0.07 and $F_{(1,25)}$ = 1.4, p = 0.26; *Figure 2D*].

Notably, our results also showed some anatomical selectivity. Indeed, when looking at CA3 region, the results showed that HFD induced an increase of c-Fos+ cells which was not attenuated by silencing either vHPC->NAc or vHPC–mPFC pathway as demonstrated by an effect of diet for either vHPC->NAc or vHPC->mPFC groups [$F_{(1,24)}$ = 15.5, p = 0.0006 and $F_{(1,25)}$ = 10.9, p = 0.0029, respectively] but no effect of DREADD or interaction [*F*'s < 2.1, p > 0.16; *Figure 2—figure supplement 1A*].

In addition, the impact of diet and DREADD was not restricted to the ventral part of the hippocampus. We indeed found that HFD increases c-Fos expression in both the NAc and the mPFC as shown in *Figure 2—figure supplement 1B,C* [diet effect: $F_{(1,22)}$ = 7.0, p = 0.015 and $F_{(1,24)}$ = 14.9, p = 0.0008, respectively]. Interestingly, DREADD was also found to decrease the amount of c-Fos+ cells in the NAc and to a lesser extent in the mPFC [$F_{(1,22)}$ = 8.6, p = 0.0076 and $F_{(1,24)}$ = 2.6, p = 0.11; interaction *F*'s ≤ 1].

**Table 1.** Body weight and fat mass measurements.
Fat mass is expressed as a percentage of body weight. Significant diet effect for both body weight and % of fat content (p < 0.003).

| Parameter/group | CD mCherry vHPC–NAc (*n* = 7) | CD mCherry vHPC–mPFC (*n* = 7) | CD Gi vHPC–NAc (*n* = 8) | CD Gi vHPC–mPFC (*n* = 9) | HFD mCherry vHPC–NAc (*n* = 8) | HFD mCherry vHPC–mPFC (*n* = 7) | HFD Gi vHPC–NAc (*n* = 22) | HFD Gi vHPC–mPFC (*n* = 22) |
|---|---|---|---|---|---|---|---|---|
| Body weight (g) | 29.3 ± 0.6 | 30.1 ± 0.9 | 30.5 ± 0.6 | 30.3 ± 0.7 | 30.4 ± 0.9 | 32.6 ± 0.8 | 31.7 ± 0.6 | 33.3 ± 0.7 |
| % of fat content | 7.7 ± 1.2 | 7.1 ± 1.6 | 9.8 ± 1.5 | 8.7 ± 1.8 | 16.8 ± 3.5 | 20.0 ± 4.9 | 20.6 ± 2.1 | 23.3 ± 2.4 |

The online version of this article includes the following source data for table 1:

**Source data 1.** Body weight and fat mass measurements.

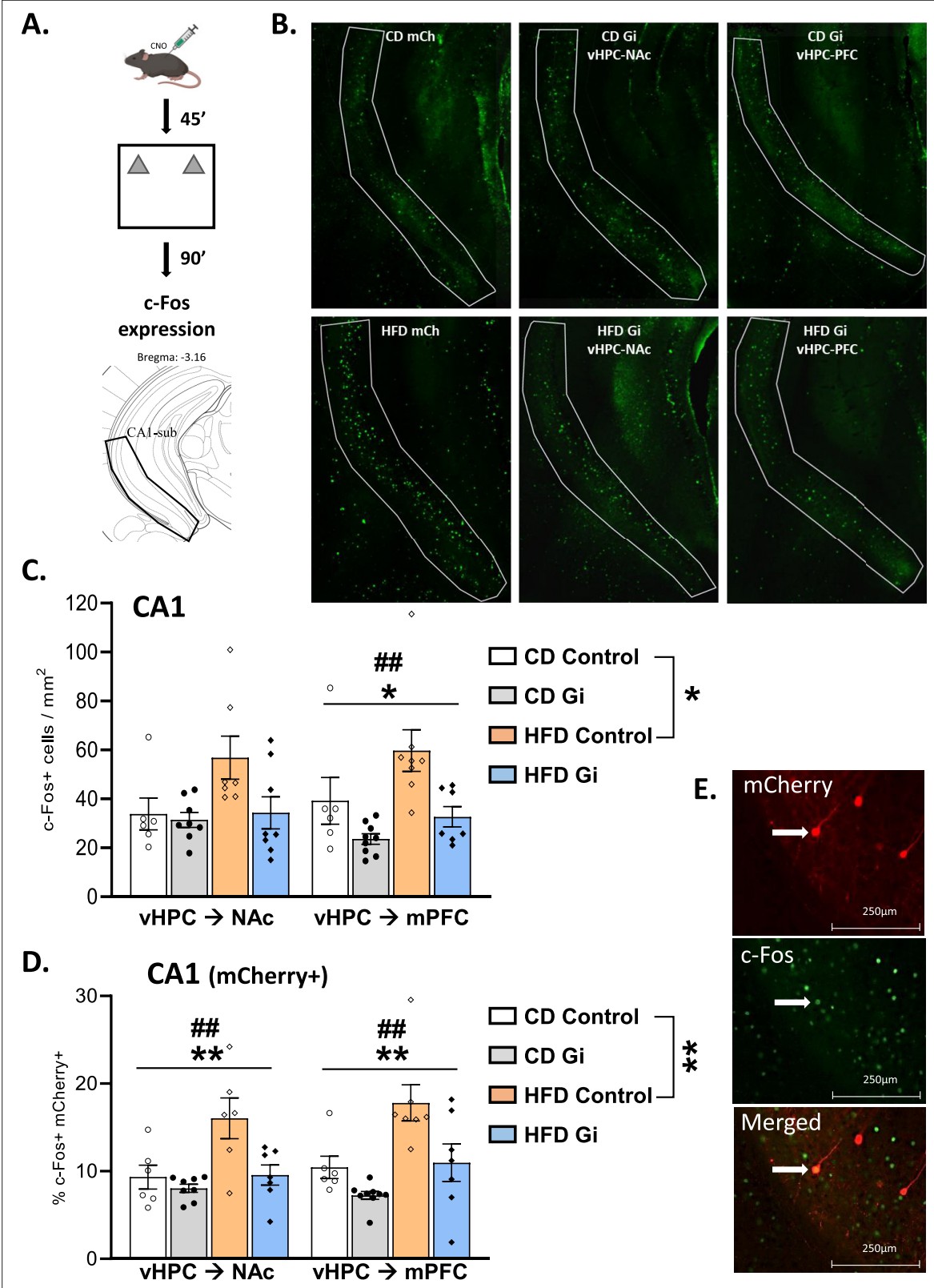

**Figure 2.** Effects of high-fat diet and chemogenetic silencing of ventral hippocampus (vHPC)–nucleus accumbens (NAc) or vHPC–medial prefrontal cortex (mPFC) pathway on c-Fos expression in ventral CA1/subiculum. (**A**) Schema of exposure to a novel arena containing two identical and novel objects. (**B**) Representative images of c-Fos-positive (Fos+) cells in the ventral CA1/subiculum for each group. (**C**) Quantification of c-Fos+ cells in the ventral CA1/subiculum of the different groups. Data are shown as the number of c-Fos+ cells per mm². The area corresponding to ventral CA1/

*Figure 2 continued on next page*

*Figure 2 continued*

subiculum area is delineated. Scale bar is set at 500 nm. (**D**) Representative example of mCherry and c-Fos labelling in the ventral CA1/subiculum. (**E**) Percentage of mCherry+ and c-Fos+ cells over total number of mCherry+ in the different groups. Scale bar is set to 250 nm and data are shown as a percentage of cells. Each point represents a single animal value. Diet effect: *p < 0.05, **p < 0.01; DREADD effect: ##p < 0.01 (two-way analysis of variance [ANOVA]).

The online version of this article includes the following source data and figure supplement(s) for figure 2:

**Source data 1.** Effects of high-fat diet and chemogenetic silencing of ventral hippocampus (vHPC)–nucleus accumbens or vHPC–medial prefrontal cortex pathway on c-Fos expression in ventral CA1/subiculum.

**Figure supplement 1.** Effect of high-fat diet and chemogenetic manipulation on c-Fos expression in hippocampus, nucleus accumbens (NAc) and medial prefrontal cortex (mPFC).

**Figure supplement 1—source data 1.** Effect of high-fat diet and chemogenetic manipulation on c-Fos expression in hippocampus, nucleus accumbens and medial prefrontal cortex.

Altogether, these results show that (1) HFD enhanced c-Fos expression in the ventral CA1–subiculum and more specifically in NAc- and mPFC-projecting vHPC neurons after object-based training and (2) silencing either vHPC->NAc or vHPC->mPFC pathway normalized HFD-induced over-activation, therefore validating our intersectional approach.

## Silencing of vHPC–NAc, but not vHPC–mPFC, pathway rescued HFD-induced long-term ORM deficits

We previously demonstrated that HFD exposure induced long-term deficits in ORM (*Naneix et al., 2021*) and that these deficits could be restored by chemogenetic silencing of the vHPC during training. We therefore wondered here whether such rescue of function could be mediated by selective silencing of either the vHPC->NAc or vHPC->mPFC pathways.

As shown in *Figure 3A*, CD Control animals spent more time exploring the new object than the familiar one, which indicates that our training procedure leads to robust long-term ORM (24 hr). Consistent with our previous study, we found that HFD exposure induced a strong memory deficit given that HFD animals spent the same amount of time exploring new and familiar objects. Such a pattern of results is supported by statistical analysis given that a two-way analysis of variance (ANOVA) with diet and pathway as factor leads to a significant effect of diet ($F_{(1,22)}$ = 10.9, p = 0.0032) but no effect of pathway or interaction ($F$'s < 0.1, p > 0.9; see *Table 3*).

We then analysed each pathway separately using two-way ANOVAs with diet and DREADD as factors. Such an analysis performed on vHPC–NAc groups leads to a lack of both diet and DREADD effects [$F$'s < 1, p > 0.3; *Figure 3A*, left] but, importantly however, there was a significant diet × DREADD interaction [$F_{(1,29)}$ = 10.2, p = 0.01]. Post hoc analyses reveal a higher preference for the novel object in HFD-Gi than in the HFD Control (p = 0.027). Looking at the performance of each group, CD Control group explored preferentially the novel object [one sample *t*-test against 50%; $t_{(6)}$ = 2.67, p = 0.037]. In contrast, there was no difference in the exploration of the two objects in HFD Control group [$t_{(5)}$ = 0.73, p = 0.49] but a significant difference in HFD-Gi group [$t_{(11)}$ = 6.65, p < 0.0001] which

**Table 2.** Object training before cellular imaging.

| Parameter/group | CD mCherry vHPC–NAc **CNO** **(n = 6)** | CD mCherry vHPC–mPFC **CNO** **(n = 6)** | CD Gi vHPC–NAc CNO (n = 8) | CD Gi vHPC–mPFC CNO (n = 9) | HFD Gi vHPC–NAc/mPFC *saline* **(n = 7)** | HFD mCherry vHPC–NAc/mPFC **CNO** **(n = 8)** | HFD Gi vHPC–NAc CNO (n = 7) | HFD Gi vHPC–mPFC CNO (n = 7) |
|---|---|---|---|---|---|---|---|---|
| Time (s) object 1 | 23.5 ± 2.9 | 20.9 ± 1.2 | 18.4 ± 2.0 | 20.2 ± 1.7 | 22.9 ± 2.3 | 23.1 ± 1.2 | 20.8 ± 2.4 | 23.1 ± 2.4 |
| Time (s) object 2 | 22.3 ± 1.8 | 21.9 ± 1.2 | 19.7 ± 2.3 | 22.1 ± 2.1 | 20.5 ± 1.0 | 22.2 ± 1.9 | 20.0 ± 2.3 | 22.1 ± 1.3 |
| Total time (s) 1 + 2 | 45.8 ± 3.0 | 42.8 ± 1.6 | 38.1 ± 2.5 | 42.3 ± 2.6 | 43.4 ± 3.1 | 45.4 ± 3.0 | 40.8 ± 4.1 | 45.3 ± 3.3 |
| mCherry+ cells (total) | 1061 | 817 | 1328 | 1153 | 1222 | 1452 | 1088 | 839 |
| mCherry+ cells/mouse | 177 ± 53 | 136 ± 32 | 166 ± 29 | 128 ± 26 | 153 ± 17 | 182 ± 37 | 155 ± 19 | 120 ± 30 |

The online version of this article includes the following source data for table 2:

**Source data 1.** Object training before cellular imaging.

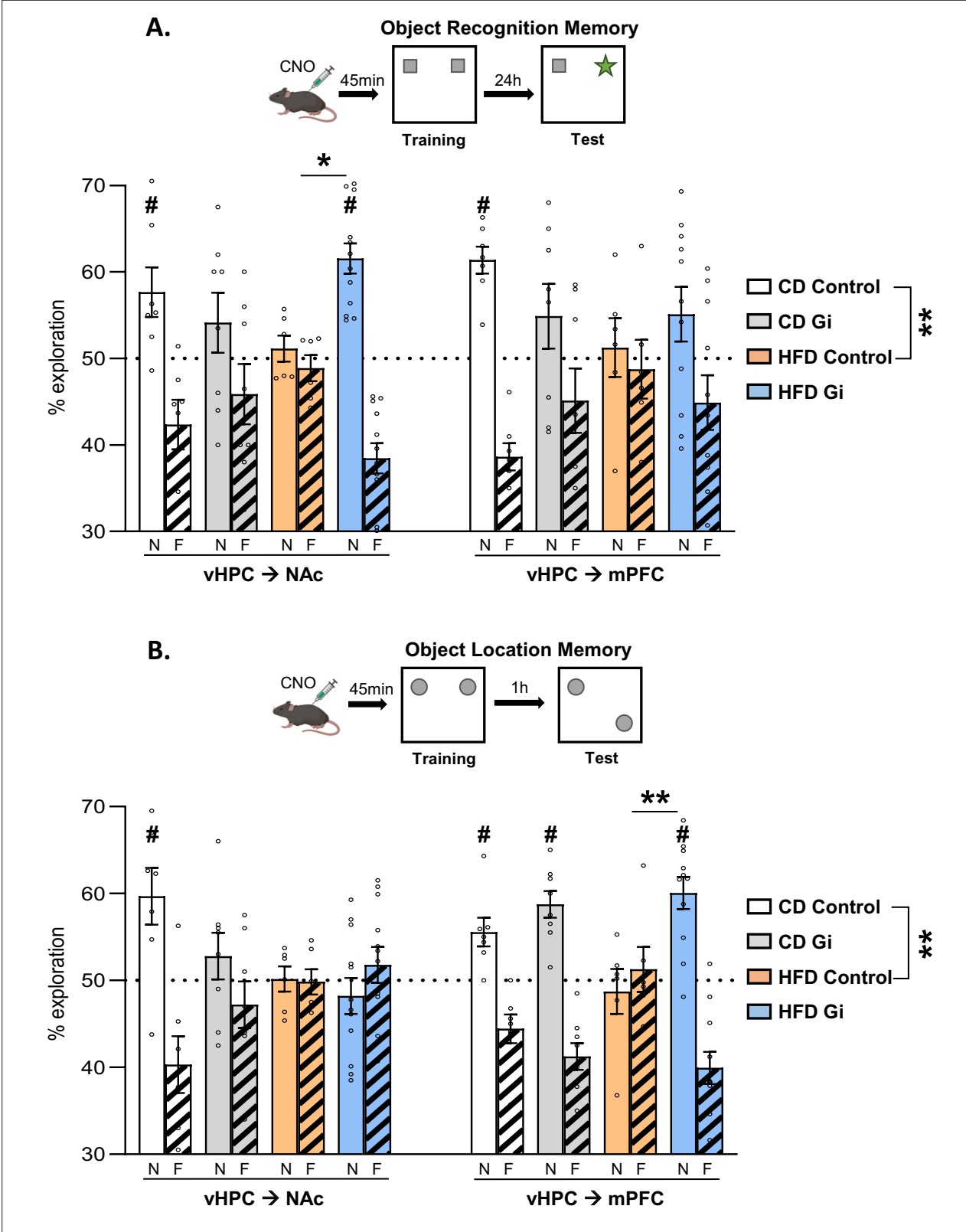

**Figure 3.** Impacts of high-fat diet and chemogenetic silencing of ventral hippocampus (vHPC)–nucleus accumbens (NAc) or vHPC–medial prefrontal cortex (mPFC) pathway on object-based memory. (**A**) Schema of object recognition memory (ORM) task (top) and ORM performance expressed as percentage of exploration time of novel (empty bars) or familiar (striped bars) object over both objects (bottom). (**B**) Schema of object location memory (OLM) task (top) and OLM performance expressed as percentage of exploration time of novel (empty bars) or familiar (striped bars) location over both

eLife Research article

Neuroscience

*Figure 3 continued*

objects (bottom). Each point represents a single animal value. Diet effect: **p < 0.01 (two-way analysis of variance [ANOVA], diet × pathway). Difference between groups: *p < 0.05, **p < 0.01 (two-way ANOVA, diet × DREADD for each pathway, significant interaction followed by post hoc test). Different from 50%: #p < 0.05 (one-sample *t*-test).

The online version of this article includes the following source data and figure supplement(s) for figure 3:

**Source data 1.** Impacts of high-fat diet and chemogenetic silencing of ventral hippocampus (vHPC)–nucleus accumbens or vHPC–medial prefrontal cortex pathway on object-based memory.

**Figure supplement 1.** Effect of high-fat diet and chemogenetic manipulation on anxiety-like behaviours.

**Figure supplement 1—source data 1.** Effect of high-fat diet and chemogenetic manipulation on anxiety-like behaviours.

indicates that inactivation of the vHPC–NAc pathway during training restored HFD-induced ORM deficits.

Similar analysis performed on the vHPC->mPFC pathway yielded to a different pattern of results. Indeed, a two-way ANOVA with diet and DREADD as factors results in an absence of both diet and DREADD effect but more importantly to a lack of diet x DREADD interaction (All *F*'s < 1.5, p > 0.2; *Figure 3A*,right). Additional analyses looking at the performance of each group show that ORM was apparent in the CD Control group [$t_{(7)}$ = 7.3, p = 0.0003] but not in the others (t < 1.6, p > 0.14).

Taken together, these results demonstrate an important functional dissociation since chemogenetic silencing of vHPC->NAc pathway but not of vHPC->mPFC pathway was shown to rescue the ORM impairment due to periadolescent HFD diet exposure.

## Silencing of vHPC–mPFC, but not vHPC–NAc, pathway rescued HFD-induced OLM deficits

We next evaluated the effect of specifically silencing the vHPC pathways on another object-based memory task, the OLM, known to be impaired by periadolescent HFD (*Glushchak et al., 2021*; *Khazen et al., 2019*; *Valladolid-Acebes et al., 2013*).

As shown in *Figure 3B*, CD Control animals spent more time exploring the displaced object than the one left in the same location, which indicates that our training procedure leads to robust memory of the object location. Consistent with published reports, our results show that that HFD exposure induced a strong OLM deficit given that HFD animals spent the same amount of time exploring the two objects. Such conclusions are supported by statistical analysis on control (no DREADD) animals which show a significant effect of diet [$F_{(1,22)}$ = 11.5, p = 0.0026] but no effect of pathway or diet × pathway interaction [*F*'s < 1.3, p > 0.26; *Figure 3B*; *Table 4*].

Silencing of vHPC->NAc or vHPC->mPFC pathway has a differential impact on the deficits in OLM resulting from HFD exposure, given that silencing of vHPC->NAc has no impact whereas silencing of

**Table 3.** Object recognition memory task.
% exploration represents the time exploring one object over total exploration time of both objects × 100.

| Parameter/group | CD mCherry vHPC–NAc (n = 7) | CD mCherry vHPC–mPFC (n = 7) | CD Gi vHPC–NAc (n = 8) | CD Gi vHPC–mPFC (n = 8) | HFD mCherry vHPC–NAc (n = 6) | HFD mCherry vHPC–mPFC (n = 6) | HFD Gi vHPC–NAc (n = 12) | HFD Gi vHPC–mPFC (n = 11) |
|---|---|---|---|---|---|---|---|---|
| ORM % exploration | 51.2 ± 2.9 | 50.9 ± 2.2 | 48.6 ± 4.4 | 50.3 ± 2.1 | 53.9 ± 2.0 | 45.9 ± 4.8 | 49.0 ± 2.7 | 50.9 ± 2.2 |
| Time (s) object to be changed | 10.2 ± 0.6 | 10.2 ± 0.4 | 9.7 ± 0.9 | 10.1 ±0.4 | 10.8 ± 0.4 | 9.2 ± 1.0 | 9.8 ± 0.5 | 10.2 ± 0.5 |
| Time (s) object not to be changed | 9.8 ± 0.6 | 9.8 ± 0.4 | 10.3 ± 0.9 | 9.9 ± 0.4 | 9.2 ± 0.4 | 10.8 ± 1.0 | 10.2 ± 0.5 | 9.8 ± 0.5 |
| Time (s) to reach criterium (training) | 400.1 ± 40.8 | 345.7 ± 42.7 | 285.6 ± 25.5 | 292.6 ± 31.9 | 395.7 ± 65.8 | 362 ± 79.8 | 298.5 ± 32.4 | 301.4 ± 34.7 |
| Time (s) to reach criterium (test) | 401.0 ± 44.6 | 292.0 ± 42.8 | 305.0 ± 26.1 | 296.6 ± 55.6 | 374.7 ± 79.6 | 343.7 ± 62.7 | 229.1 ± 20.6 | 285.7 ± 26.1 |

The online version of this article includes the following source data for table 3:

**Source data 1.** Object recognition memory task.

**Table 4.** Object location memory task.

% exploration represents the time exploring one object over total exploration time of both objects × 100.

| Parameter/group | CD mCherry vHPC–NAc (n = 7) | CD mCherry vHPC–mPFC (n = 7) | CD Gi vHPC–NAc (n = 8) | CD Gi vHPC–mPFC (n = 8) | HFD mCherry vHPC–NAc (n = 6) | HFD mCherry vHPC–mPFC (n = 6) | HFD Gi vHPC–NAc (n = 12) | HFD Gi vHPC–mPFC (n = 11) |
|---|---|---|---|---|---|---|---|---|
| OLM % exploration | 45.1 ± 1.1 | 48.1 ± 2.1 | 47.0 ± 1.6 | 50.6 ± 2.6 | 49.1 ± 1.0 | 47.05 ± 2.2 | 50.6 ± 1.5 | 47.7 ± 1.9 |
| Time (s) object to be displaced | 9.0 ± 0.2 | 9.6 ± 0.4 | 9.4 ± 0.3 | 10.1 ± 0.5 | 9.9 ± 0.1 | 9.5 ± 0.5 | 10.1 ± 0.3 | 9.5 ± 0.4 |
| Time (s) object not to be displaced | 11.0 ± 0.2 | 10.4 ± 0.4 | 10.6 ± 0.3 | 9.9 ± 0.5 | 10.1 ± 0.1 | 10.5 ± 0.5 | 9.9 ± 0.3 | 10.5 ± 0.4 |
| Time (s) to reach criterium (training) | 390.6 ± 59.9 | 279.3 ± 43.0 | 349.6 ± 40.1 | 316.0 ± 57.4 | 351.7 ± 28.0 | 327.8 ± 42.6 | 340.8 ± 37.3 | 358.4 ± 24.7 |
| Time (s) to reach criterium (test) | 407.9 ± 55.2 | 327.3 ± 30.6 | 343.8 ± 44.7 | 341.1 ± 45.7 | 292.7 ± 52.3 | 330.0 ± 44.9 | 337.9 ± 27.8 | 339.6 ± 36.9 |

The online version of this article includes the following source data for table 4:

**Source data 1.** Object location memory task.

vHPC->mPFC was efficient in restoring near to normal memory performance. Such conclusions are supported by statistical analysis. Indeed, there was a significant diet × DREADD interaction when vHPC–mPFC pathway was manipulated [$F_{(1,28)}$ = 4.3, p = 0.047; diet effect: $F_{(1,28)}$ = 1.9, p = 0.17; DREADD effect: $F_{(1,28)}$ = 13.6, p = 0.001; *Figure 3B*, right] but not when vHPC–NAc pathway was silenced [$F_{(1,29)}$ = 0.9, p = 0.34; diet effect: $F_{(1,29)}$ = 7.6, p = 0.01; DREADD effect: $F_{(1,29)}$ = 3.0, p = 0.09; *Figure 3B*, left]. Post hoc analyses indicated a significantly higher preference for the displaced object in HFD Gi vHPC–mPFC group than in HFD Control vHPC–mPFC group (p = 0.0017).

Looking at the performance of each group, CD-fed Control mice explored preferentially the displaced object [one sample *t*-test against 50%: vHPC–NAc: $t_{(6)}$ = 2.97, p = 0.025; vHPC–mPFC: $t_{(6)}$ = 3.36, p = 0.015] whereas HFD Control groups exhibited OLM deficits, exploring both objects similarly [vHPC–NAc: $t_{(5)}$ = 0.12, p = 0.91; vHPC–mPFC: $t_{(5)}$ = 0.49, p = 0.64; *Figure 3B*]. In HFD-fed mice, chemogenetic silencing of the vHPC–NAc pathway before training had no effect on OLM performance [$t_{(11)}$ = 0.9, p = 0.40], whereas manipulation of the vHPC–mPFC pathway improved it [$t_{(10)}$ = 5.4, p = 0.0003]. In CD-fed mice, chemogenetic silencing of the vHPC–NAc pathway before training impaired OLM performance [$t_{(7)}$ = 1.0, p = 0.33] whereas manipulation of the vHPC–mPFC pathway had no effect [$t_{(7)}$ = 5.7, p = 0.0007].

Taken as a whole, these results demonstrate that silencing of the vHPC->mPFC, but not the vHPC->NAc, pathway restored HFD-induced OLM deficits.

## HFD and projecting vHPC cells manipulation did not affect anxiety-like behaviour

Adult HFD intake (*Fulton et al., 2022*) or manipulation of specific vHPC pathways (*Padilla-Coreano et al., 2016*; *Parfitt et al., 2017*) has been reported to affect anxiety-like behaviour. We therefore evaluated, whether anxiety was affected in our conditions using the Elevated-Plus Maze test (*Figure 3—figure supplement 1A*).

There was no effect of HFD or vHPC pathway inhibition on the percentage of time spent in open arm [diet × pathway: $F_{(1,17)}$ = 0.7, p = 0.41; diet × DREADD: $F_{(1,25)}$ = 1.1, p = 0.29 for vHPC–NAc groups, $F_{(1,24)}$ = 0.6, p = 0.44 for vHPC–mPFC groups; *Figure 3—figure supplement 1B*] or the percentage of open arm entries [diet × pathway: $F_{(1,17)}$ = 0.2, p = 0.69; diet × DREADD: $F_{(1,25)}$ = 0.01, p = 0.92 for vHPC–NAc groups, $F_{(1,24)}$ = 0.18, p = 0.68 for vHPC–mPFC groups; *Figure 3—figure supplement 1C*].

These results indicate that neither periadolescent HFD nor vHPC pathway manipulation affect anxiety-like behaviour, and therefore exclude that any effects on memory performance would be due to differences in anxiety levels among the groups.

## Discussion

The current study provides evidence that periadolescent HFD-induced deficits in non-aversive and non-rewarded object-based memory in mice, without affecting anxiety-like behaviour. These memory

deficits were associated with enhanced activation of vHPC neurons projecting to NAc and mPFC. Using a pathway-specific chemogenetic approach that was effective in normalizing this activation, we demonstrated that (1) silencing of NAc-projecting, but not mPFC-projecting, vHPC neurons restored long-term ORM deficits induced by HFD whereas (2) inactivation of mPFC-projecting, but not NAc-projecting, vHPC neurons rescued HFD-induced OLM deficits. Our results therefore revealed a double dissociation in the beneficial outcomes of vHPC–NAc or vHPC–mPFC silencing on HFD-induced ORM and OLM deficits, without any effect on anxiety-like behaviour.

Our previous studies revealed that 12 weeks of periadolescent HFD induced a higher c-Fos activation of the dorsal hippocampus after object-based training (*Biyong et al., 2021*). Here, we obtained similar c-Fos overexpression in the CA1/subiculum of the vHPC after object training, as well as in projecting areas NAc shell and mPFC, providing evidence that a large neural network is overactive in HFD-fed mice following object training. In HPC, HFD effect is more pronounced in ventral CA1 and subiculum, which represent the main output of the vHPC (*Britt et al., 2012*; *Cenquizca and Swanson, 2007*; *Ciocchi et al., 2015*; *Gergues et al., 2020*; *Liu and Carter, 2018*). This could be related to HFD-induced morphological and electrophysiological changes in CA1 as it was reported that 8–12 weeks of post-weaning HFD intake enhances dendritic spine density in CA1 pyramidal neurons (*Valladolid-Acebes et al., 2013*) and induces aberrant in vivo long-term potentiation in CA1 (*Vouimba et al., 2021*). Thanks to our intersectional virus approach, we were able to restrict our c-Fos analyses to NAc- or mPFC-projecting vHPC neurons located in ventral CA1/subiculum. Periadolescent HFD induced an increased percentage of c-Fos expressing neurons in both pathways after object training, corroborating what was found at the level of the whole ventral CA1–subiculum. Importantly, Clozapine-*N*-Oxide (CNO) decreased the activation of vHPC-to-NAc and vHPC-to-mPFC neurons expressing inhibitory DREADDs in HFD-fed mice, validating the efficacy of our intersectional chemogenetic approach. This restricted chemogenetic inhibition tends to attenuate HFD-induced overactivation of the whole ventral CA1/subiculum, possibly through a network effect.

Our results indicate that periadolescent HFD induces long-term ORM deficits, corroborating our recent studies showing specific HFD effects on long-term, but not short-term, ORM (*Biyong et al., 2021*; *Naneix et al., 2021*). We had previously reported that chemogenetic inactivation of vHPC projecting neurons during training, by restricting DREADDs expression primarily to excitatory principal neurons thanks to the use of a CaMKII promoter, is able to restore HFD-induced ORM deficits (*Naneix et al., 2021*). Here, we obtained similar ORM improvement in HFD-fed animals using inactivation of NAc-projecting vHPC, but not mPFC-projecting vHPC, neurons during memory formation. This finding supports previous results indicating that HFD affects NAc functions after periadolescent HFD exposure (*Ducrocq et al., 2019*; *Labouesse et al., 2013*; *Naneix et al., 2017*). Moreover, silencing NAc-projecting vHPC neurons attenuated ORM in controls in line with previous findings demonstrating the role of NAc in ORM consolidation (*Sargolini et al., 2003*) and in integrating HPC-derived information during memory consolidation (*Kerfoot and Williams, 2018*; *Roozendaal et al., 2001*).

Previous findings indicate that periadolescent exposure to HFD drastically affects spatial memory assessed in the Morris water maze or the radial arm maze (*Boitard et al., 2012*; *Boitard et al., 2014*) but also using non-aversive and non-rewarded OLM (*Glushchak et al., 2021*; *Khazen et al., 2019*; *Valladolid-Acebes et al., 2013*). We here demonstrate that silencing of the vHPC–mPFC, but not the vHPC–NAc, pathway restores OLM deficits induced by periadolescent HFD. Our results are related to previous studies indicating that periadolescent HFD alters mPFC functions (*Labouesse et al., 2017*; *Reichelt et al., 2019*; *Yaseen et al., 2019*) and that vHPC–mPFC pathway modulates different aspects of object-based memory (*Barker et al., 2017*; *Morici et al., 2022*; *Nelson et al., 2011*). It must be stressed that OLM was tested 1 hr after training. As CNO action lasts several hours (*Alexander et al., 2009*), mPFC–vHPC pathways were inactivated during both OLM training and test. Additional experiments will determine whether chemogenetic inactivation of vHPC–mPFC pathways after training and before test still has a beneficial effect on OLM performance therefore supporting mPFC role in orchestrating action to perform a given task (*Friedman and Robbins, 2022*).

Recent studies indicate that a very small proportion (5–10%) of vHPC neurons project to two or three areas (*Ciocchi et al., 2015*; *Gergues et al., 2020*; *Parfitt et al., 2017*). However, the double dissociation in our effects of pathway-specific silencing on ORM and OLM precludes that collaterals of mPFC- and NAc-projecting vHPC neurons may be involved in our effects. Also, this dissociation observed in the present study excludes any influences of sensory, motor, motivational or attentional

processes, which would have similarly affected both memory tasks. Finally, our diet or pathway manipulation did not affect anxiety-like behaviour, ruling out that our effects on memory were due to differences in anxiety levels among the groups. This is in line with previous studies performed after juvenile or periadolescent HFD exposure (*André et al., 2014*; *Boitard et al., 2015*; *Hayes et al., 2023*; *Khazen et al., 2019*; *Tsan et al., 2022*) but in contrast with others showing that adult HFD exposure triggered anxiety-like behaviour (*Décarie-Spain et al., 2018*; *Fulton et al., 2022*). This suggests a developmental HFD impact on anxiety which deserves more investigation.

## Conclusions

The current study provides evidence that periadolescent HFD induces deficits in different types of memory through an over-activation of specific vHPC efferent pathways and dampening this activation alleviates these memory deficits. These findings extend our knowledge about the cerebral impact of obesogenic diet focusing on brain network and connectivity and emphasize the distinct role of specific vHPC efferent pathways in different types of object-based memory. Future studies are required to examine the effects of manipulation of vHPC pathways, including those to anterior olfactory nucleus (*Aqrabawi and Kim, 2018*), in other types of memory impairments promoted by HFD, such as social and olfactory memory (*Reichelt et al., 2019*; *Yaseen et al., 2019*). Moreover, further investigation is also necessary to decipher what is the origin of this HPC over-activation in obesogenic diet-fed rodents.

According to the important role played by HPC hyperactivity in memory decline associated with normal and pathological ageing (*Bakker et al., 2012*; *Yassa et al., 2011*) as well as disruption of vHPC efferent pathways in neuropsychiatric disorders (*Bagot et al., 2015*; *Li et al., 2015*; *Phillips et al., 2019*), future research has to contribute to the development of novel therapeutic strategies to alleviate disorders characterised by impaired HPC homeostasis.

## Materials and methods
### Animals, diet, and housing

Male C57BL/6J mice aged of 3 weeks old (Janvier Labs, France) were divided randomly into groups of 8 per cage (45 × 25 × 20 cm, containing a paper house, nesting material and a small wooden stick) and had ad libitum access to a control diet (CD; $n$ = 43; 2.9 kcal/g; 8% lipids, 19% proteins, 73% carbohydrates mostly from starch; A04, SAFE) or a HFD ($n$ = 64; 4.7 kcal/g; 45% lipids mostly saturated fat from lard, 20% proteins, 35% carbohydrates mainly from sucrose; D12451, Research Diet). We focused on males given that female rodents do not consistently exhibit memory deficits after post-weaning obesogenic diet intake (see *Abbott et al., 2016*; *Hwang et al., 2010*), which is probably related to ovarian hormones that appear to protect females from obesity and metabolic impairments (*Palmer and Clegg, 2015*). All animals were housed in a temperature-controlled room (22 ± 1°C) maintained under a 12-hr light/dark cycle (lights on at 8:00 am, lights off at 8:00 pm) and had free access to food and water during 12 weeks (before the beginning of behaviour) as well as during all behavioural procedures before euthanasia (13–14 weeks of diet exposure in total). Animals were weighed at arrival, before and after surgery, before each behavioural test as well as before euthanasia. All animal care and experimental procedures were in accordance with the INRAE Quality Reference System and with both French (Directive 87/148, Ministère de l'Agriculture et de la Pêche) and European legislations (Directive 86/609/EEC). They followed ethical protocols approved by the Region Aquitaine Veterinary Services (Direction Départementale de la Protection des Animaux, approval ID: B33-063-920) and by the animal ethic committee of Bordeaux CEEA50. Every effort was made to minimize suffering and reduce the number of animals used. Both surgeries and behavioural experiments were performed at adulthood. The day before euthanasia, fat mass (in grams) was measured by nuclear magnetic resonance (minispec LF90 II, Bruker, Wissembourg, 67166) (*Naneix et al., 2021*) and divided by body weight (in grams) to express the percentage of fat content for each mouse. Three homozygous Ai14 Cre reporter adult mice [B6.Cg-Gt(ROSA)$^{26Sortm14(CAG-tdTomato)Hze}$/J or simply Ai14(RCL-tdT)-D; Jackson laboratory] under CD were also used to characterize vHPC-to-NAc and vHPC-to-mPFC pathways allowing the Cre-dependent expression of the red fluorophore td-Tomato specifically in vHPC projecting neurons and their efferents (see below).

## Viral vectors and drugs

An AAV carrying the Cre recombinase was injected in vHPC for the experiment performed in the Ai14(RCL-tdT)-D mice (AAV1-CaMKII-Cre). For the chemogenetic manipulation of specific pathways, an anterograde AAV carrying Cre-dependent inhibitory DREADD (AAV8-hSyn1-dlox-hM4DGi-mCherry or Gi) or control virus (AAV8-hSyn1-dlox-mCherry) was injected in vHPC in combination with a retrograde canine virus (CAV2) carrying the Cre recombinase (CAV2-Cre) injected in either the mPFC or the NAc. This led to 8 groups [2 diets x 2 viruses injected in vHPC × 2 target areas (NAc or PFC) injected with retrograde virus]: CD Control vHPC–NAc, CD Control vHPC–mPFC, CD Gi vHPC–NAc, CD Gi vHPC–mPFC; HFD Control vHPC–NAc, HFD Control vHPC–mPFC, HFD Gi vHPC–NAc, and HFD Gi vHPC–mPFC.

The exogenous DREADD ligand CNO was dissolved in 0.9% saline for a final concentration of 2 mg/kg. Saline solution was used for vehicle injections. Both CNO and vehicle were freshly prepared every day and delivered by intraperitoneal injection (10 ml/kg) 45 min before the training phases of the object memory tests (Recognition and Location) and before the beginning of the elevated plus-maze (EPM) experiment.

## Stereotaxic surgery

After 7–8 weeks under CD or HFD, mice were anaesthetized under isoflurane (5% induction; 1–2% maintenance), injected with the analgesic buprenorphine (Buprecar, 0.05 mg/kg s.c.) and the non-steroidal anti-inflammatory drug carproxifen (Carprox, 5 mg/kg s.c.) and were placed on a stereotaxic apparatus (David Kopf Instruments). The scalp was shaved, cleaned, and locally anaesthetized with a local subcutaneous injection of lidocaine (Lurocaine, 0.1 ml). Viral vectors were infused using a 10-µl Hamilton syringe (Hamilton) and an ultra-micro pump (UMP3, World Precision Instruments, USA). For the experiment performed in the Ai14(RCL-tdT)-D mice, two injections per vHPC (AP −3.2 mm, ML ±3.2 mm from Bregma, DV −3 and −4 mm from the skull surface, according to *Paxinos and Franklin, 2004*) of 1 µl each were performed with the vector AAV1-CaMKII-Cre. For the dual virus experiments, 1 µl of the AAV (AAV-dlox-hM4Di or AAV control) was injected over 6 min (150 nl/min) in the vHPC at 1 site in each hemisphere, coordinates were AP −3.2 mm, ML ±3.2 mm from Bregma, DV −4 mm from the skull surface. Then 250 nl in mPFC (AP +1.9 mm, ML ±0.3 mm, DV −3 mm from skull) or 400 nl in NAc shell (AP +1.2 mm, ML ±0.6 mm, DV −4.8 mm from skull) of CAV2 virus were injected at a rate of 100 nl/min over 2 min 30 s and 4 min, respectively. In all cases, the pipette was left in place for a 5-min diffusion period, before being slowly removed. The incision was closed with sutures and the animal was kept on a heating pad until recovery. Mice were single housed for 4 days and their body weight and behaviour were closely monitored. Then, they were housed in groups of 4 mice per cage and 4 weeks later (allowing optimal virus expression) behavioural tests started.

## Behavioural procedures

The eight groups were submitted to different object-based memory tasks known to be impaired by periadolescent HFD, namely long-term ORM (*Biyong et al., 2021*; *Naneix et al., 2021*) and short-term OLM (*Glushchak et al., 2021*; *Khazen et al., 2019*; *Valladolid-Acebes et al., 2013*). Anxiety-like behaviours were also evaluated in an EPM. The order of tests was performed in a random way. Animals were handled twice a day for 3 days before the beginning of the first behavioural test. All behavioural tests were performed during light-phase and under white light (15 lux). Between each trial, arena (and objects where appropriate) were cleaned with 10% of alcohol. All behaviour analysis was scored online, apart from the EPM that was automatically analysed by SMART system (Bioseb, Vitrolles, France).

### Object recognition memory

During training, two identical new objects were presented in a novel open field arena (40 × 40 × 40 cm, wood) and each mouse was left freely to explore them. Long-term memory was assessed 24 hr later, by randomly replacing one of the objects by a novel one. In both phases, object exploration, defined as nose and whiskers pointed towards the object in a distance of less than 1–1.5 cm away, was manually quantified by a trained experimenter blind to experimental groups. In both phases, the session duration was 10 min max but the mouse was removed from the apparatus after 20 s of total exploration for both objects which reduces inter-individual variability (*Leger et al., 2013*), otherwise mice were excluded

from analysis. Data were presented as the percentage of exploration of novel (or familiar) object calculated as time exploring novel (or familiar) object over total exploration time of both objects × 100.

### Object location memory

Two identical new objects (different from ORM) were presented during training in a new room with a new open field arena (40 × 40 × 40 cm, plastic) and each mouse was left freely to explore them. Short-term memory was assessed 11 hr later, by randomly placing one of the objects in a novel location. In both phases, an inclusion criterion of 20 s of total exploration was set at a 10-min exploration maximum, otherwise mice were excluded from analysis. Data were presented as the percentage of exploration of novel (or familiar) location calculated as time exploring novel (or familiar) location on total exploration time of both objects × 100.

### Elevated plus-maze

Mice were allowed to freely explore the plus-shaped acrylic maze sized 30 × 8 × 15 cm (closed arms) and 30 × 8 cm (open arms) connected by a central part (8 × 8 cm) for 10 min. The maze is elevated 120 cm above the floor. A mouse was considered to enter one zone only when it placed all four limbs in any particular part of the maze. Time spent in the open versus the closed arms was recorded and results are depicted as a percentage of open arm time [calculated as time in open arms over total time spent in both open and closed arms (with exclusion of time spent in the central area) × 100]. In addition, a percentage of open arm entries was also calculated with respect to the total number of entries in both open and closed arms. An increased percentage for the open arms (time and/or entries) indicates low anxiety.

## Tissue collection and immunohistochemistry

All mice were sacrificed after behavioural testing. For experiments evaluating c-Fos activation, four to eight mice in each of the eight groups were injected with CNO 45 min before being placed in new open field arena (40 × 40 × 40 cm) with two identical new objects (corresponding to ORM and OLM training) during 8 min and were sacrificed 90 min after [3 mice presenting very low number of mCherry labelled cells (<30) in vHPC were excluded; see *Table 2*]. Moreover, additional HFD Gi mice (3 vHPC–NAc and 4 vHPC–mPFC) were injected with saline (instead of CNO) to control for CNO effect (*Table 2*). As similar levels of Fos+ cells were found between these HFD Gi-saline and HFD mCherry-CNO groups, indicating an absence of CNO effect by itself on c-Fos expression, they were merged (vHPC–NAc $n = 7$ + vHPC–mPFC $n = 8$) and appears as HFD Control groups on *Figure 2* and *Figure 2—figure supplement 1*.

Animals were deeply anaesthetized with a mix of pentobarbital and lidocaine (Exagon, 300 mg/kg and Lurocaine, 30 mg/kg) before being transcardiacally perfused with phosphate-buffered saline (PBS) solution followed by 4% paraformaldehyde (Sigma-Aldrich). Brains were collected, post-fixed in 4% paraformaldehyde at 4°C for 2 days, then switched to PBS solution and stored at 4°C until slicing. 40 μm coronal sections were cut via vibratome (Leica) and stored in cryoprotective solution (glycerol and ethylene glycol Sigma-Aldrich in PBS) at −20°C.

## DREADD expression

On the first day, slices were washed with PBS and incubated with 0.33% $H_2O_2$ solution in PBS for 30 min. Then, they were incubated with blocking solution [0.2% Triton, 3% foetal bovine serum (FBS), in PBS] for 90 min followed by an incubation with a rabbit anti-DsRed antibody (1:2000, in blocking solution) overnight at 4°C. Next, slices were incubated with a biotinylated goat anti-rabbit antibody (1:1000 in PBS with 1% FBS) for 90 min at room temperature, followed by an 1-hr incubation in avidin–biotin–peroxidase solution (Vectastain). Slices were then washed with PBS and Tris buffer solution (TBS, pH = 7.4) followed by diaminobenzidine (DAB) incubation for 15–30 min (1 tablet of DAB and 1 tablet $H_2O_2$; 5 ml of distilled $H_2O$ in 20 ml of TBS). After stopping the reaction, slices were stored at 4°C. Slices were then mounted on gelatin-coated slides, covered by medium (Southern Biotech) and cover-slipped. Each section was photographed (Nikon-ACT-1 software).

## Pathway activation

A double immunofluorescence was performed. Slices were washed with PBS solution, incubated with blocking solution for 90 min then with a combination of two primary antibodies: rabbit anti-Fos

(1:2000) and chicken anti-mCherry (1:5000), all in blocking solution of 3% FBS, 0.2% Triton in PBS (72 hr, at 4°C). Then, slices were incubated with a combination of two secondary antibodies, goat anti-chicken (1:1000, A488) and goat anti-rabbit (1:1000, A594). Slices were stored at 4°C until mounting on non-gelatin-coated slides followed by cover of DAPI fluoromount (Southern Biotech) and cover-slipped. All slices were photographed with Nanozoomer slide scanner Hamamatsu NANOZOOMER 2.0HT (Bordeaux Imaging Center, Univeristy of Bordeaux, France). QuPath program QuPath v.0.3.0 (*Bankhead et al., 2017*) was used for quantification of c-Fos-positive cells in hippocampus (CA1 and CA3), NAc shell, and mPFC.

## Statistics

Two mice died after surgery (one CD and one HFD) and five mice were excluded after histological control (one CD and four HFD with unilateral labelling). Data were analyzed with Prism Software (GraphPad) and are expressed as mean ± mSEM. Comparison between groups was realized using three-way ANOVA for weight and fat mass (diet × pathway × DREADD) and two-way ANOVA for c-Fos and behaviours (either diet × pathway in Control mice or diet × DREADD for each pathway manipulation) followed when the interaction was significant by Tukey's post hoc analysis. Moreover, for each group, ORM and OLM performances were compared against 50% exploration ratio (chance level) using one-sample $t$-test. Significance was set at $p \leq 0.05$.

## Acknowledgements

The authors thank Gregory Artaxet and Eva Bruchet (NutriNeuro lab) for taking daily care of the animals, Léa Décarie-Spain for help during surgery, Lola Fauré for some illustrations and the Bordeaux Imaging Center (a service unit of University of Bordeaux and a national infrastructure, France BioImaging) where part of microscopy was completed. This work was supported by INRAE (to GF), CNRS (to EC), and French National Research Agency (ANR-14-CE13-0014 GOAL and ANR-15-CE17-0013 OBETEEN to EC and GF and ANR-16-CE37-0010 ORUPS to GF). IB was the recipient of a PhD fellowship from the French Ministry of Research and Higher Education (2018–2021).

## Additional information

### Funding

| Funder | Grant reference number | Author |
| --- | --- | --- |
| Agence Nationale de la Recherche | ANR-14-CE13-0014 GOAL | Etienne Coutureau |
| Agence Nationale de la Recherche | ANR-15-CE17-0013 OBETEEN | Etienne Coutureau |
| Agence Nationale de la Recherche | ANR-16-CE37-0010 ORUPS | Guillaume Ferreira |

The funders had no role in study design, data collection, and interpretation, or the decision to submit the work for publication.

### Author contributions

Ioannis Bakoyiannis, Conceptualization, Formal analysis, Investigation, Writing – original draft; Eva Gunnel Ducourneau, Conceptualization, Formal analysis, Investigation, Writing – review and editing; Mateo N'diaye, Alice Fermigier, Celine Ducroix-Crepy, Investigation; Clementine Bosch-Bouju, Funding acquisition; Etienne Coutureau, Pierre Trifilieff, Writing – review and editing; Guillaume Ferreira, Conceptualization, Formal analysis, Supervision, Funding acquisition, Writing – original draft, Writing – review and editing

### Author ORCIDs

Ioannis Bakoyiannis ⑩ http://orcid.org/0000-0002-6324-7321
Eva Gunnel Ducourneau ⑩ http://orcid.org/0000-0002-9254-4405
Etienne Coutureau ⑩ http://orcid.org/0000-0001-6695-020X

Guillaume Ferreira https://orcid.org/0000-0001-5984-8143

**Ethics**

All animal care and experimental procedures were in accordance with the INRAE Quality Reference System and with both French (Directive 87/148, Ministx00E7;re de l'Agriculture et de la Px00EA;che) and European legislations (Directive 86/609/EEC). They followed ethical protocols approved by the Region Aquitaine Veterinary Services (Direction DU+00E9; partementale de la Protection des Animaux, approval ID: B33-063-920) and by the animal ethic committee of Bordeaux CEEA50.

**Decision letter and Author response**

Decision letter https://doi.org/10.7554/eLife.80388.sa1

Author response https://doi.org/10.7554/eLife.80388.sa2

## Additional files

**Supplementary files**
• MDAR checklist

**Data availability**

All data generated during this study are available in the associated supplementary files.

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
