## [Editor Report]

This valuable study explores the neuronal pathways through which adolescent diet influences later memory processes. The data convincingly show that obesogenic high-fat diet during adolescence impairs distinct forms of memory via distinct ventral hippocampal projections to nucleus accumbens v. medial prefrontal cortex. This study will be of interest to those interested in development, diet, metabolism, learning and memory, and the intersection of these factors.

---

## [Decision Letter]

**Decision letter after peer review:**

Thank you for submitting your article "Obesogenic diet induces circuit-specific memory deficits in mice" for consideration by *eLife*. Your article has been reviewed by 3 peer reviewers, and the evaluation has been overseen by a Reviewing Editor and Kate Wassum as the Senior Editor. The following individual involved in the review of your submission has agreed to reveal their identity: Jessica L Ables (Reviewer #1).

Essential revisions:

1) It is not clear how long the animals spend with the objects in either the training or testing phase. The phrase "an inclusion criteria of 20 s of total exploration was set at a 10 minutes exploration max" is not clear. Is 10 min the duration of the phases?

2) A one-way ANOVA was used to compare the groups in figures 2 and 3 and in Supplemental figures 2 and 3. However, a one-way ANOVA is used when there is one variable, and in this case, there are multiple variables: diet, CNO treatment, and circuit. The authors should use a more appropriate test or provide a rationale for why they chose one-way ANOVA.

3) The light/dark assay is another relatively simple test that also assesses whether an animal will explore a novel and exposed/brightly lit area. Concordance of the results of both measures of anxiety would strengthen the conclusion that anxiety is not a confounding factor.

4) The intersectional strategy used to specifically manipulate vHPC cells projecting to either NAc and mPFC allow them to restrict the manipulation to a selected subpopulation of hippocampal cells. While this approach is very convincing, a control is needed for changes in neuronal activity due to optogenetic manipulation in the rest of the hippocampal neurons and in downstream regions (NAc and mPFC). This control is important to build up the hypothesis that the 2 pathways are selectively recruited in the memory tasks.

5) A downstream effect of the optogenetic manipulation on NAc and mPFC cellular populations should be shown to support claims that the chemogenetic inhibition decreases the activation of the pathway and not only of vHPC projecting neurons.

6) Why cFos data in NAc and PFC are not shown, as stated in the discussion? They represent an important control and should be part of the main results of the paper.

7) it is unclear how the pathways end up diverging for memory if they do not show any differences in cfos during training. Perhaps there are pathway-specific differences in cfos following the ORM and OLM tests? It is difficult to support the claim that there are pathway differences in memory following inactivation if we do not see any pathway-specific change in activity.

8) Optogenetic control groups (m4d saline and control virus CNO) are well designed and should be presented separately even if the behavioral outcome is the same since they are controlling for different effects.

9) There needs to be a HFD- / DREADD+ control for Figures 2 and 3 to control for the effect of DREADDs on normal chow, irrespective of HFD.

*Reviewer #1 (Recommendations for the authors):*

1) It is not clear how long the animals spend with the objects in either the training or testing phase. The phrase "an inclusion criteria of 20 s of total exploration was set at a 10 minutes exploration max" is not clear. Is 10 min the duration of the phases?

2) A one-way ANOVA was used to compare the groups in figures 2 and 3 and in Supplemental figures 2 and 3. However, a one-way ANOVA is used when there is one variable, and in this case, there are multiple variables: diet, CNO treatment, and circuit. The authors should use a more appropriate test or provide a rationale for why they chose one-way ANOVA.

3) The light/dark assay is another relatively simple test that also assesses whether an animal will explore a novel and exposed/brightly lit area. Concordance of the results of both measures of anxiety would strengthen the conclusion that anxiety is not a confounding factor.

*Reviewer #2 (Recommendations for the authors):*

This manuscript from Bakoyiannis and colleagues brings new results to unravel the role of vHPC selected pathways in memory processes and relates it to the development of memory deficits due to high-fat diet and obesity. By manipulating hippocampal circuits, the authors suggested different pathways involved in different types of memory.

– The intersectional strategy the authors used to specifically manipulate vHPC cells projecting to either NAc and mPFC allows them to restrict the manipulation to a selected subpopulation of hippocampal cells. While this approach is very convincing, authors should control for changes in neuronal activity due to optogenetic manipulation in the rest of the hippocampal neurons and in downstream regions (NAc and mPFC). This control is important to build up the hypothesis that the 2 pathways are selectively recruited in the memory tasks.

– Why cFos data in NAc and PFC are not shown, as stated in the discussion? They represent an important control and should be part of the main results of the paper.

– Optogenetic control groups (m4d saline and control virus CNO) are well designed and should be presented separately even if the behavioural outcome is the same since they are controlling for different effects.

*Reviewer #3 (Recommendations for the authors):*

It would help if the authors would analyze the data between NAcc and mPFC separately for the control mice, which might reveal pathway differences.

The authors should test for pathway-specific differences in cfos by examining cfos following OLM and ORM tests.

The authors should separate mPFC and NAcc controls and conduct the analyses for each pathway separately. They should also add an HFD -, DREADD + control and then conduct a two-way ANOVA for each pathway.

---

## [Author Response]

Essential revisions:1) It is not clear how long the animals spend with the objects in either the training or testing phase. The phrase "an inclusion criteria of 20 s of total exploration was set at a 10 minutes exploration max" is not clear. Is 10 min the duration of the phases?

We apologize for lacking clarity on this point. Our protocol for object-based memory tasks was based on a previous paper which standardized the procedure in mice (Leger et al., Nature Protocols, 2013). This protocol reduces inter-individual variability with the use of a selection criterion based on 20 seconds of exploration for both objects during each session (training and test). The session duration was 10min max but the mouse was removed from the apparatus after 20 seconds of total exploration for both objects. Table 3 and 4 report the time to reach this criterium during training and test for ORM (Table 3, page 21 lines 673-675) and OLM (Table 4, page 22 lines 675-7.) tasks which did not reveal any difference between the groups during both training and test sessions. We more clearly describe the protocol for both ORM and OLM training and test sessions on page 11 lines 359361.

2) A one-way ANOVA was used to compare the groups in figures 2 and 3 and in Supplemental figures 2 and 3. However, a one-way ANOVA is used when there is one variable, and in this case, there are multiple variables: diet, CNO treatment, and circuit. The authors should use a more appropriate test or provide a rationale for why they chose one-way ANOVA.

We fully agree with this criticism pointed out by the 3 Reviewers. To perform two-way ANOVA (allowing to evaluate the global effect of diet, DREADD or pathway and their interactions), we conducted new experiments in order to add new groups such as control diet (CD)-fed animals with inhibitory DREADD in the ventral Hippocampus-Nucleus Accumbens (CD vHPC-NAc Gi) or ventral Hippocampus-medial Prefrontal cortex (CD vHPC-mPFC Gi) pathways and to increase the number of animals in pre-existing groups.

This led to 8 groups as indicated in the Results and Material and Methods sections (page 3 lines 9294 and page 10 lines 318-321): 4 vHPC-NAc groups (CD vHPC-NAc mCherry, CD vHPC-NAc Gi, HFD vHPC-NAc mCherry, HFD vHPC-NAc Gi) and 4 vHPC-mPFC groups (CD vHPC-mPFC mCherry, CD vHPCmPFC Gi, HFD vHPC-mPFC mCherry, HFD vHPC-mPFC Gi).

For c-Fos and behavioral experiments, this allowed us to perform a two-way ANOVA to evaluate first the effects of diet X pathways in Control groups (comparing the 4 mCherry groups: CD vHPC-NAc mCherry, CD vHPC-mPFC mCherry, HFD vHPC-NAc mCherry, HFD vHPC-mPFC mCherry) and then the effects of diet x DREADD for each pathway manipulation [comparing the 4 vHPC-NAc groups on one hand and the 4 vHPC-PFC groups on the other hand].

These new analyses, indicated in the Material and Methods section (page 13 lines 416-418), are now described in the Results section and revealed a general diet effect and general DREADD effect for cFos results (page 4 lines 103-124, new Figure 2 and new Figure 2—figure supplement 1), diet X DREADD interactions in ORM (after vHPC-NAc pathway manipulation specifically; page 5 lines 137-154, new Figure 3A) and OLM tasks (after vHPC-mPFC pathway manipulation only; page 6 lines 168187 new Figure 3B) and no effect in anxiety-like behavior (page 7 lines 197-201; new Figure 3—figure supplement 1).

3) The light/dark assay is another relatively simple test that also assesses whether an animal will explore a novel and exposed/brightly lit area. Concordance of the results of both measures of anxiety would strengthen the conclusion that anxiety is not a confounding factor.

We don’t think anxiety is a confounding factor in our results. The effect of HFD exposure were previously evaluated on different anxiety-like behaviors such as EPM, Open-Field, novelty-suppressed feeding or light/dark box (see for review Fulton et al., 2022). An important factor seems to be the developmental period of HFD exposure. Whereas 2 to 3 months of HFD intake at adulthood, starting when rodents are 8-12 weeks old, enhances anxiety-like behaviors (see for instance our recent paper: Demers et al., 2020 and for review Fulton et al., 2022), HFD exposure during adolescence (starting at weaning, e.g. when rodents are 3 weeks old) does not affect anxiety-like behaviors as we assessed using EPM or Open-Field tests (Boitard et al., 2015; Khazen et al., 2019; Andre et al., 2014 and Figure 3—figure supplement 1) or others did using other tests to evaluate anxiety-like behaviors (Hayes et al., 223; Tsan et al., 2022). We now precised in the Results section that “adult” HFD intake has been reported to affect anxiety-like behavior (page 7 lines 193-194) and added 3 new references in the Discussion section indicating the absence of effect of adolescent HFD on anxiety-like behaviors (Andre et al., 2014; Hayes et al., 223; Tsan et al., 2022, page 9 lines 266-267 and in the References section: page 14 lines 439-441, page 15 lines 506-509 and page 18 lines 592-594).

Moreover, we would highlight the fact that we found in HFD-fed mice a double dissociation in the effects of vHPC-NAc and vHPC-PFC pathways manipulation on ORM and OLM performance, respectively. This clearly excludes anxiety as a confounding factor to explain the beneficial effects of pathway manipulation on memory. Indeed, if pathway manipulation would improve memory performance by improving anxiety-like behavior, this should affect indifferently both memory tasks.

4) The intersectional strategy used to specifically manipulate vHPC cells projecting to either NAc and mPFC allow them to restrict the manipulation to a selected subpopulation of hippocampal cells. While this approach is very convincing, a control is needed for changes in neuronal activity due to optogenetic manipulation in the rest of the hippocampal neurons and in downstream regions (NAc and mPFC). This control is important to build up the hypothesis that the 2 pathways are selectively recruited in the memory tasks.5) A downstream effect of the optogenetic manipulation on NAc and mPFC cellular populations should be shown to support claims that the chemogenetic inhibition decreases the activation of the pathway and not only of vHPC projecting neurons.6) Why cFos data in NAc and PFC are not shown, as stated in the discussion? They represent an important control and should be part of the main results of the paper.

To address these 3 points, we performed new experiments and new c-Fos analyses.

First, in ventral hippocampus, our c-Fos results showed some anatomical selectivity. Silencing either vHPC->NAc or vHPC-mPFC pathway normalized HFD-induced over-activation in CA1 but not in CA3, therefore revealing that we manipulate specific hippocampal subpopulation (see page 4 lines 103119, new Figure 2D and new Figure 2—figure supplement 1A).

Second, we also showed diet and DREADD effects in downstream regions. c-Fos expression in the NAc of vHPC-NAc groups revealed a diet effect, with higher c-Fos expression in HFD-fed groups, and a DREADD effect, with lower expression in Gi groups (page 4 lines 121-124; New Figure 2—figure supplement 1B). Similarly, c-Fos expression in the mPFC of vHPC-mPFC groups was higher in HFD-fed groups (diet effect) and tended to be lower in Gi groups (page 4 lines 121-124; Figure 2—figure supplement 1C).

7) it is unclear how the pathways end up diverging for memory if they do not show any differences in cfos during training. Perhaps there are pathway-specific differences in cfos following the ORM and OLM tests? It is difficult to support the claim that there are pathway differences in memory following inactivation if we do not see any pathway-specific change in activity.

We apologize for lacking clarity on this point. ORM and OLM training sessions are based on the same protocol: exploration of two copies of the same novel object placed on two adjacent corners of an open-field environment. The mice therefore learn during this training session (at least) 2 things: the features of the object (allowing them to subsequently recognize and differentiate this “now familiar” object from a novel one, e.g. ORM) and the location of these objects (to recognize if one of the objects was displaced to a different corner, e.g. OLM). If we consider that vHPC-NAc overactivation is responsible for HFD-induced ORM deficit and vHPC-PFC overactivation is responsible for HFD-induced OLM deficit, it is therefore not surprising that we did not see any pathway-specific change in activity after training as both pathways were overactivated. Pathway-specific differences are only revealed when each pathway is manipulated separately.

We focused on training, instead of test, for c-fos assessment as we demonstrated that HFD induces long-term, but not short-term, ORM deficits indicating an effect on post-training consolidation memory process rather than on retrieval (Naneix et al., 2021).

8) Optogenetic control groups (m4d saline and control virus CNO) are well designed and should be presented separately even if the behavioral outcome is the same since they are controlling for different effects.

We have now provided information about HFD Gi-saline and HFD mCherry-CNO groups in the new Table 2 (page 21 lines 671-672). Moreover, we indicated in the Material and Methods section (page 12 lines 384-387): “additional HFD Gi mice (3 vHPC-NAc and 4 vHPC-mPFC) were injected with saline (instead of CNO) to control for CNO effect (Table 2). As similar levels of Fos+ cells were found between these HFD Gi-saline and HFD mCherry-CNO groups, indicating an absence of CNO effect by itself on cFos expression, they were merged (vHPC-NAc n=7 and vHPC-mPFC n = 8) and appears as HFD Control groups on (Figure 2 and Figure 2—figure supplement 1)”.

9) There needs to be a HFD- / DREADD+ control for Figures 2 and 3 to control for the effect of DREADDs on normal chow, irrespective of HFD.

As indicated for point 2, we conducted new experiments to add new groups on normal chow (CD) with inhibitory DREADD in the ventral Hippocampus-Nucleus Accumbens (CD vHPC-NAc Gi) or ventral Hippocampus-Prefrontal cortex (CD vHPC-PFC Gi) pathways. Both CD DREADD groups showed decrease Fos expression in each pathway relative to CD mCherry groups (see new Figure 2) and attenuated performance for ORM (new Figure 3A) whereas only CD vHPC-NAc Gi group showed impaired OLM (see Figure 3B). This additional CD groups allowed us to perform a two-way ANOVA between all the groups to evaluate the effects of diet (CD vs HFD), DREADD (mCherry vs Gi), pathway (vHPC-NAc vs vHPC-PFC) and their interactions. These new analyses, indicated in the Material and Methods section (page 13 lines 416-418), are now described in the Results section and revealed a general Diet effect and general DREADD effect for c-Fos results (page 4 lines 102-124, new Figure 2 and new Figure 2—figure supplement 1), diet X DREADD interactions in ORM (after vHPC-NAc pathway manipulation specifically; page 5 lines 137-154, new Figure 3A) and OLM tasks (after vHPCmPFC pathway manipulation only; page 6 lines 168-187 new Figure 3B) and no effect in anxiety-like behavior (page 7 lines 197-201; new Figure 3—figure supplement 1).

Reviewer #1 (Recommendations for the authors):1) It is not clear how long the animals spend with the objects in either the training or testing phase. The phrase "an inclusion criteria of 20 s of total exploration was set at a 10 minutes exploration max" is not clear. Is 10 min the duration of the phases?

We thank the Reviewer for this comment. We replied to this query above (please see point 1 of “Essential revisions” section).

2) A one-way ANOVA was used to compare the groups in figures 2 and 3 and in Supplemental figures 2 and 3. However, a one-way ANOVA is used when there is one variable, and in this case, there are multiple variables: diet, CNO treatment, and circuit. The authors should use a more appropriate test or provide a rationale for why they chose one-way ANOVA.

Based on this comment we added new groups and modified statistical analyses accordingly. We have corrected this in the manuscript (please see point 2 of “Essential revisions” section).

3) The light/dark assay is another relatively simple test that also assesses whether an animal will explore a novel and exposed/brightly lit area. Concordance of the results of both measures of anxiety would strengthen the conclusion that anxiety is not a confounding factor.

We thank the Reviewer for this comment. Please see our answer above (point 3 of “Essential revisions” section).

Reviewer #2 (Recommendations for the authors):This manuscript from Bakoyiannis and colleagues brings new results to unravel the role of vHPC selected pathways in memory processes and relates it to the development of memory deficits due to high-fat diet and obesity. By manipulating hippocampal circuits, the authors suggested different pathways involved in different types of memory.– The intersectional strategy the authors used to specifically manipulate vHPC cells projecting to either NAc and mPFC allows them to restrict the manipulation to a selected subpopulation of hippocampal cells. While this approach is very convincing, authors should control for changes in neuronal activity due to optogenetic manipulation in the rest of the hippocampal neurons and in downstream regions (NAc and mPFC). This control is important to build up the hypothesis that the 2 pathways are selectively recruited in the memory tasks.– Why cFos data in NAc and PFC are not shown, as stated in the discussion? They represent an important control and should be part of the main results of the paper.

To address these points, we performed new c-Fos experiments. Please see our response above (points 4-5-6 in the “Essential revisions” section).

– Optogenetic control groups (m4d saline and control virus CNO) are well designed and should be presented separately even if the behavioural outcome is the same since they are controlling for different effects.

Thank you for this comment (we used chemogenetic, instead of optogenetic, approach in our study though). Please see our response to point 8 in the “Essential revisions” section above.

Reviewer #3 (Recommendations for the authors):It would help if the authors would analyze the data between NAcc and mPFC separately for the control mice, which might reveal pathway differences.

Thank you for this comment. We have answered this question in the “Essential revisions” section above. Please see points 2, 4-5-6 and 9.

The authors should test for pathway-specific differences in cfos by examining cfos following OLM and ORM tests.

We thank the Reviewer for this comment. Please see our answer to point 7 in the “Essential revisions” section above.

The authors should separate mPFC and NAcc controls and conduct the analyses for each pathway separately. They should also add an HFD -, DREADD + control and then conduct a two-way ANOVA for each pathway.

We thank the Reviewer for this comment. Please see our answers to points 2, 4-5-6 and 9 in the “Essential revisions” section and new Figure 2, new Figure 3, new Figure 2—figure supplement 1 and new Figure 3—figure supplement 1.